# Evaluation of Acute and Subacute Toxicity and LC-MS/MS Compositional Alkaloid Determination of the Hydroethanolic Extract of *Dysphania ambrosioides* (L.) Mosyakin and Clemants Flowers

**DOI:** 10.3390/toxins14070475

**Published:** 2022-07-12

**Authors:** Fahd Kandsi, Fatima Zahra Lafdil, Amine Elbouzidi, Saliha Bouknana, Achraf Miry, Mohamed Addi, Raffaele Conte, Christophe Hano, Nadia Gseyra

**Affiliations:** 1Laboratory of Bioresources, Biotechnology, Ethnopharmacology and Health, Faculty of Sciences, Mohammed First University, B.P. 717, Oujda 60000, Morocco; kandsi.fahd@ump.ac.ma (F.K.); lafdil.fatimazahra@ump.ac.ma (F.Z.L.); bouknana.saliha@gmail.com (S.B.); ngseyra@hotmail.com (N.G.); 2Laboratoire d’Amélioration des Productions Agricoles, Biotechnologies’ et Environnement (LAPABE), Faculté des Sciences, Université Mohammed Premier, Oujda 60040, Morocco; amine.elbouzidi@ump.ac.ma; 3Pathology Department, Mohammed VI University Hospital, Oujda 60040, Morocco; achrafmiry@outlook.com; 4Research Institute on Terrestrial Ecosystems (IRET)—CNR, Via Pietro Castellano 111, 80131 Naples, Italy; raffaele.conte86@tiscali.it; 5Laboratoire de Biologie des Ligneux et des Grandes Cultures, INRAE USC1328, University of Orleans, CEDEX 2, 45067 Orléans, France

**Keywords:** *Dysphania ambrosioides*, Mexican tea, alkaloids, LC-MS/MS, acute toxicity, subacute toxicity, in silico, ADMET analysis

## Abstract

*Dysphania ambrosioides* (L.) Mosyakin and Clemants is a medicinal plant that has traditionally been used to cure a range of diseases. There has been no thorough investigation of the potential toxicity of this plant. The objective of this study is to assess the acute and subacute toxicity of *D. ambrosioides* hydroethanolic extract (DAHE), as well as it alkaloids composition, utilizing LC-MS/MS analysis. An in silico approach was applied to determine pharmacokinetic parameters and to predict the toxicity of *D. ambrosioides* identified alkaloids. A 14-day treatment with a single oral dose of 1–7 g/kg was carried out to investigate acute toxicity. DAHE was given orally at dosages of 5, 50, and 500 mg/kg for 15 days in the subacute toxicity investigation, and body weight and biochemical parameters were evaluated. Livers, kidneys, lungs, and heart were examined histologically. Chromatographic investigation revealed the existence of nine alkaloids, with N-formylnorgalanthamine being the most prevalent. The oral LD_50_ value of DAHE was found to be 5000 mg/kg in an acute toxicity study. No variations were observed with respect to food intake, water consumption, mortality, or body and organ weight in the subacute toxicity study. On the other hand, DAHE (500 mg/kg) significantly enhanced alanineaminotransferase, aspartate aminotransferase, and urea. Liver and kidney histological examinations revealed modest infiltration of hepatocyte trabeculae by inflammatory cells in the liver and slight alteration in the kidney histoarchitecture. According to our findings, DAHE exhibits low to moderate toxicity.

## 1. Introduction

Medicinal plants have been employed as ethnic treatments for numerous ailments since prehistoric times. They have been widely considered a possible source of new phytochemicals with bioactive properties. These phytomedicines can lead to drug discovery or become a prospective drug in order to treat a number of ailments. As a result, it is critical to establish and quantify the bioactive potential of ethnomedicinal plants by focusing on phytochemistry and developing them as a source of therapeutic agents [1,2,3].

Herbal medicine and natural remedies are an essential part of Moroccan cultural heritage [4,5]; they play a significant role in Moroccans’ daily lives [5,6], and they are used to cure a vast range of illnesses [7]. Although these plants have shown extraordinary potential phytotherapeutic qualities and are in high demand globally, there are still questions about their use and safety [8]. As a result, a number of studies are being carried out to determine the toxicity of therapeutic plants and their compounds. Toxicity is the state of being poisonous, describing the undesirable consequences caused by toxicants interacting with cells. This method of action can vary, depending on the cell membrane and the chemical characteristics of the toxicants. Toxicity can occur inside the cell membrane, on the cell surface, below the cell membrane, or in the extracellular matrix. Toxins injure key organs, including the liver and kidney, in the vast majority of instances [9].

*Dysphania ambrosioides* (L.) Mosyakin and Clemants belongs to the Chenopodiaceae family and is locally known in Morocco as “M’Khinza”. Several ethnobotanical studies have revealed that the indigenous Moroccan people employed *D. ambrosioides* to treat a variety of diseases, including fever, female infertility, rheumatism, sexual impotence [10], digestive disorders, nervous system problems, and respiratory issues [11,12]. It is also used in traditional medicine for its antifungal, antiaflatoxigenic, and pesticide properties [13,14]. Results of the phytochemical screening of genus Chenopodium have shown the presence of substances such as primary metabolites (carbohydrates, proteins, amino acids, non-polar components, hormones, and lipids) and secondary metabolites (flavonoids, saponins, sterols, terpenes, and alkaloids) [15,16,17].

According to pharmacological studies [18], this plant has antibacterial and antifungal effects, as well as insecticidal characteristics [19,20], antioxidant properties [16,21,22], anti-inflammatory properties [21,22], and myorelaxant and antispasmodic qualities [16]. Furthermore, this plant is used as a vermifuge to cure gastrointestinal disorders and to help pregnant people [23]. It was also revealed that *D. ambrosioides* might be used to treat Alzheimer’s disease [24].

The possible toxicity of this plant has not been thoroughly investigated before. Thus, we intended to evaluate the acute and subacute toxicity of hydroethanolic extract of *D. ambrosioides*, as in previous research, where hydroethanolic extraction was conducted [25,26]. Alkaloid composition was determined utilizing LC-MS/MS analysis.

## 2. Results

### 2.1. Phytochemical Analysis

Table 1 shows the different alkaloid components of the hydroethanolic extract of *Dysphania ambrosioides*. Phytochemical analysis of the hydroethanolic extract reveals the presence of N-formylnorgalanthamine alkaloids, followed by trisphaeridine, galanthamine, crinine, demethylmaritidine, anhydrolycorine, nor-galanthamine and nor-galanthamine, and then peramine and ergovaline (Appendix A, Table 1, Appendix A).

### 2.2. ADME Analysis and Toxicity Prediction

It is generally recognized that poor pharmacokinetic properties (absorption, distribution, metabolism, excretion, and toxicity) can undermine good pharmacological activity. Furthermore, undesirable pharmacokinetics and toxicity are significant reasons for the costly failure of drug discovery in the clinical phase. To determine whether *D. ambrosioides* ethanolic crude extract is a good candidate for a viable medication, ADMET characteristics were evaluated using in silico techniques (Table 2).

ADMET analysis was carried out in order to predict a number of parameters corresponding to each alkaloid, including pharmacokinetics and toxicity, evaluating its potential as a drug candidate. Lipinski’s rule of five predicts whether a molecule is orally bioavailable by responding to some criteria (MLOGP < 4.15, H-bond donors < 5, H-bond acceptors < 10, MW < 500, N or O < 10) [32]. All the alkaloids satisfy Lipinski’s rule of five, except ergovaline, which exhibits one violation by slightly exceeding the maximal required molecular weight (MW > 500). Therefore, bioavailability scores are set at 0.55% for molecules that satisfy the RO5 (Rule of Five).

Water solubility can be an important factor affecting absorption. According to log scale, good water solubility ranges from −4 log mol/L to 0 log mol/L [33]. All the alkaloids are water-soluble compounds.

A Caco-2 permeability prediction model (given as logP in 10–6 cm/s) is used to identify absorption and drug transport mechanisms [34]. Besides **8** and **9**, all the compounds display good Caco-2 permeability.

Gastrointestinal (GI) absorption and blood–brain barrier (BBB) permeability of all identified substances were determined using the brain or intestinal estimate permeation (BOILED-Egg) model (Figure 1). The results showed that all identified alkaloidal substances have a sufficient percentage of intestinal absorption according to fitting in the white area. The yolk (yellow area) cluster molecules able to pass through the BBB; therefore, with the exception of peramine **(8)** and ergovaline **(9)**, all identified alkaloids are projected to have good blood–brain barrier permeability.

P-glycoprotein (P-gp), the most important member of the ABC transporters (ATP-binding cassette transporters), is used to estimate active efflux through biological membranes and to protect the central nervous system (CNS) from xenobiotic compounds. All investigated compounds are categorized as a substrate of P-gp, besides molecule **8**, which prevents it from penetrating the CNS. Although none of the molecules showed inhibition properties with respect to P-glycoprotein I and II, ergovaline **(9)** was found to be an inhibitor of P-glycoprotein I. The volume of distribution at steady state or VDss (log L/kg) is considered low for alkaloids **1**, **5**, **7**, and **8**, whereas the other alkaloids display good distribution through plasma.

Regarding metabolism, human cytochrome P450 (CYP) isoforms (CYP2D6 and CYP3A4) that are involved in drug metabolism in the liver were also examined. Within the human body, CYP3A4 and CYP2D6 are the most therapeutically suitable drug-metabolizing enzymes [35]. Inhibiting these enzymes may cause drug toxicity, drug–drug interactions, and other side effects; it is therefore necessary to assess whether the molecules are inhibitors or non-inhibitors of the two main isoenzymes. With the exception of peramine, all investigated alkaloids are classified as substrate to CYP3A4, whereas alkaloids **3** and **4** are substrates to CYP3A4. Only anhydrolycorine and peramine were found to be non-inhibitors of CYP2D6, whereas all alkaloids are predicted to be non-inhibitors of CYP3A4, with the exception of trisphaeridine.

The Total clearance results obtained in this investigation (expressed as log mL/min/kg) indicate a good half-life. Furthermore, galanthamine **(2)** is the only substrate to renal OCT2 (organic cation transporter 2).

None of the alkaloids is linked to skin sensitization. However, alkaloids **1**, **2**, **3**, and **6** are hepatotoxic and may lead to a disruption in terms of liver function, whereas trisphaeridine, crinine, anhydrolycorine, and peramine are AMES-positive, which may causecarcinogenicactivity. Environmental toxicity of each alkaloid was also evaluated, and all investigated alkaloids were found to be toxic in a *Tetrahymen pyriformis* test, although only trisphaeridine showed high acute toxicity in flathead minnows. The maximal recommended tolerated dose of alkaloids is considered low.

According to the OECD 423 model, LD_50_ values indicated no oral acute toxicity and were labelled as class VI (‘non-toxic’) according to the Oral Toxicity Classification.

### 2.3. Acute Toxicity of DAHE in Mice

The effect DAHE on acute toxicity in mice was evaluated after a single oral administration. Loss of movement and anorexia were recorded at doses of 3 and 5 g/kg. Mortality was observed in mice at a dose of 7 g/kg. After anesthesia and dissection of the mice, a normal aspect of the organs (liver, right kidney, left kidney, and lung) was noted. Statistical analysis showed that there was no significant difference between organ weights in mice treated with DAHE and control mice (Table 3; Figure 2).

### 2.4. Subacute Toxicity

#### 2.4.1. Body Weight, Food Intake, and Water Consumption

DAHE did not cause any evident subacute toxicity at any of the doses used, nor toxicity or death in any of the treated rats. Furthermore, rats treated subacutely with repeated oral treatments of the DAHE (5, 50 or 500 mg/kg) showed no significant changes in urinary volume or food and water consumption (Table 4). Normal and treated rats appeared showed no symptoms of toxicity at the end of the study and throughout the 15-day period. According to the results shown in Figure 3, when compared to the control group, the body weights of mice in the treatment groups with dosages up to 500 mg/kg did not change substantially (*p* ˂ 0.05) during the investigation period.

#### 2.4.2. Organ Weight

Table 5 shows the weights (absolute and relative) of organs collected from normal rats and rats treated for 15 days. There was no significant difference in absolute and relative weight of the liver, kidneys, heart, and lungs between control and treatment groups.

#### 2.4.3. Biochemical Parameters

Figure 4 shows the effect of subacute DAHE therapy on aspartate aminotransferase (AST) and alanine aminotransferase (ALT) levels in rats. The biochemical markers AST and ALT, which were examined in this investigation, increased substantially in the 500 mg/kg group compared to the control group (*p* < 0.01).

With respect to the effect of subacute treatment of rats with DAHE on total cholesterol and triglycerides, the results (Figure 5) suggested that the groups treated subacutely with DAHE at concentrations of 5, 50, and 500 mg/kg experienced no significant changes in triglycerides and total cholesterol relative to the control group. In addition, the mean cholesterol values for all rats after 15-day treatment did not differ significantly between untreated and treated groups with DAHE at different doses (Figure 5).

There were no significant differences in terms of plasma urea, uric acid, and creatinine in rats following subacute DAHE administration in any of the treated groups (5, 50, and 500 mg/kg) compared to the control group after 15 days of treatment (Figure 6). Furthermore, when compared to the control group, the group treated with 500 mg/kg had a substantial increase in urea and creatinine levels (*p* < 0.001; *p* < 0.05).

The effects of subacute treatment of rats with DAHE on albumin, total bilirubin, and total protein 15 days after oral administration are shown in Figure 7. No significant changes were detected in terms of total protein levels, bilirubin, oralbumin levels in rats treated with5, 50,or 500 mg/kg of DAHE (*p* ˂ 0.05).

There were also no significant changes in plasma glycemia levels after 15 days of treatment in any of the treated groups (5, 50, and 500 mg/kg) compared to the control group (Figure 8).

#### 2.4.4. Histopathological Changes

##### Liver Histopathology

Light micrographs of liver slices of the treatment groups are presented in Figure 9: (A) control rats, (B) rats treated with 5 mg/kg DAHE, (C) rats treated with 50 mg/kg DAHE, and (D) rats treated with 500 mg/kg DAHE. Liver samples from group (D) showed mild infiltration of hepatocytes trabeculae by inflammatory cells, mostly granulocytes and lymphocytes (shown by red arrows). These findings are in line with biochemical measures, which indicated no change in AST or ALT, and no necrosis was observed.

##### Kidney Histopathology

The distal tubule, proximal tubule, and glomerulus of the kidney tissues were also investigated for any changes. The control group’s kidney histological sections were normal (glomeruli, tubules, inter stitium, and blood vessels) (Figure 10A′), where as those of the group treated subacutely with 5 and 50 mg/kg doses of DAHE (Figure 10B′) indicated a modest abnormality in the histoarchitecture of the kidneys (including a decrease in glomerulus cells) and a secondary expansion of the Bowman space (red arrow) (black star). Tubular constructions were unaffected (See Figure 10C′,D′, for example). These findings were also corroborated by the findings with respect to biochemical parameters (urea and creatinine), which showed no changes.

##### Histopathology of the Heart

The histopathology of the heart is presented in Figure 11: (A″) untreated rats, (B″) rats treated with 5 mg/kg DAHE, (C″) rats treated 50 mg/kg DAHE, and (D″) rats treated with 500 mg/kg DAHE. There were no histopathological abnormalities in any of the four rat groups.

##### Lung Histopathology

Light micrographs of different treatment groups’ lung slices. The treatment groups are represented by the numbers on the photos (Figure 12). (A‴) Untreated rats, (B‴) rats given treated with 5 mg/kg DAHE, (C‴) rats treated with 50 mg/kg DAHE, and (D‴) rats treated with 500 mg/k g DAHE. No histological abnormalities were observed in any of the four rat groups.

## 3. Discussion

The oral LD_50_ value of DAHE was 5000 mg/kg in this investigation, per the acute toxicity evaluation. According to Hodge and Sterner (2005), LD_50_ determination can be used to classify toxicity into six categories: “Class 1 = extreme toxicity, LD_50_ 1 mg/kg; Class 2 = high toxicity, LD_50_ 1–50 mg/kg; Class 3 = moderate toxicity, LD_50_ 50–500 mg/kg; Class 4 = low or mild toxicity, LD_50_ 500–5000 mg/kg; Class 5 = practically non-toxic, LD_50_ 5000–15,000 mg/kg; Class 6 = relatively harmless, LD_50_ > 15,000 mg/kg” [36]. In light of this classification and the LD_50_ value, DAHE with an LD_50_ of 5000 mg/kg might be categorized as low-toxicity or mildly hazardous.

All of the animals used in this investigation were responsive and reacted favorably to stimuli following subacute exposure. There were no deaths or clinical symptoms of local or systemically harmful consequences. The animals’ behavior was observed on a daily basis, and no changes were noted [37]. In general, a rise or reduction in body weight of animals has been considered a symptom of a negative side effects of a drug or chemical [38]. Relative organ weight also indicates whether an organ has been damaged. Organs that have been injured are susceptible to abnormal atrophy [39]. The body weights and relative organ weights of all treated rats did not differ significantly from those of the control group (*p* > 0.05). This suggests that the extract had no effect on animal appetite or growth.

Similarly, the weight of the object did not significantly alter the heart, liver, lungs, or kidneys, indicating that subacute oral administration of DAHE had no influence on normal growth. The relative organ-weighing protocol in toxicity studies its sensitivity in predicting toxicity is taken into account and is well correlated with histopathological changes.

Liver and kidneys are vital organs in the body; one is responsible for digestion and waste elimination, whereas the other is responsible for waste elimination alone [40,41]. It is necessary to know the state of liver and kidneys in order to assess the toxicity of any new drug, which can be validated through biochemical estimation [41]. The levels of two enzymes (ALT and AST) in the blood are routinely utilized as clinical biochemical markers of liver disease [42,43]. When the serum levels of ALT and AST in the 5 and 50 mg/kg groups were compared to those of the control group, they were found to be comparable. The subacute treatment of rats with a higher dose (500 mg/kg) of DAHE produced a considerable elevation in ALT and AST values (*p* < 0.01). Furthermore, when rats treated with DAHE at a dose of 500 mg/kg were compared to rats in the control group, we observed a significant rise in urea (*p* < 0.001) and creatinine (*p* < 0.05). In contrast to the control group, the DAHE (5, 50, and 500 mg/kg) groups exhibited no significant differences in triglyceride and total cholesterol levels. Based on the biochemical data, it was feasible to predict that a high dose of DAHE could cause toxicity to essential organs in rats. Finally, both groups treated with 500 mg/kg experienced minor changes after subacute DAHE administration. These alterations may be related to disturbances in the kidneys and liver. DAHE administration was not deadly, and it did not cause a hazardous change in the therapeutic dose, implying that using this product at the recommended dose is safe [44].

After chronic experimental treatment, some plant compounds can cause systemic toxicity in animals, which can manifest as weight loss, behavioral changes, and biochemical changes. Plant chemicals that cause liver and kidney disease generate toxic issues [44]. Therefore, the scientific knowledge required for an acute oral toxicity study is very necessary not only to help determine the range and concentration of doses that could be used in the future but also to reveal any clinical indications that may be caused by the chemicals under investigation.

Medicinal herbs have been used to cure a variety of disorders for hundreds of years [7]. Evaluation and assessment of the hazardous properties of an extract from a natural product is usually the initial stage in determining the pharmacological activity of a natural substance. Although, a considerable amount of information is available regarding the positive pharmacological properties of *D. ambrosioides*, there is a scarcity of knowledge about this herb’s harmful effects. Therefore, the purpose of this research was to evaluate the acute and subacute toxicity of *D. ambrosioides* hydroethanolic flower extract in mice and rats.

Phytochemical analysis by LC-MS/MS revealed the presence of nine alkaloids in the hydroethanolic extracts: trisphaeridine, galanthamine, crinine, demethylmaritidine, anhydrolycorine, nor-galanthamine, N-formylnorgalanthamine, peramine, and ergovaline. Several studies have identified the presence of alkaloids, but no study has identified these alkaloid types [45]. Sterols, saponins, and phenolic compounds have all been reported as secondary metabolites of this plant. A recent study demonstrated that *D. ambrosioides* contains alkaloids [45]. This finding correlates with our results. Another study identified a single alkaloid (1-Piperoylpiperidine) [23].

Owing to their diverse biological actions, these isoquinoline alkaloids could be a promising source of novel medications. The most important alkaloid is galantamine, which has been approved by the Food and Drug Administration (FDA) for the treatment of mild to moderate Alzheimer’s disease in research trials (AD) [46] because of its possible acetylcholine esterase-inhibiting activity [47].

## 4. Materials and Methods

### 4.1. Plant Material and Preparation of the Hydroethanolic Extract of D. ambrosioides

*Dysphania ambrosioides* (L.) Mosyakin and Clemants flowers were collected in February 2021 near “Guercif” (Eastern Morocco). A voucher specimen was placed at the Department of Biology of the Faculty of Sciences, University Mohammed the first (Oujda, Morocco) under collection number HUMPOM44.

Dried and ground flowers of *D. ambrosioides* were subjected to maceration by ethanol (70%) with continuous agitation for a week. The soluble part of this mixture was separated from the dry part, followed by filtration. In order to remove the ethanol, the obtained filtrate was subjected to reduced-pressure evaporation using a rotary evaporator. The extract was dried in the oven overnight at 40 °C and then stored at −4 °C until use.

### 4.2. LC-MS/MS Analysis of Hydroethanolic Extract

The following extraction procedures were applied to aliquots of samples (80 mg): 1 mL of ethanol was added to the aliquot. The Eppendorf was vortexed and incubated for 60 min at 45 °C in a sonicator bath. A Shimadzu ultra-high-performance liquid chromatograph (Nexera XR LC 40) was paired with an MS/MS detector for qualitative analysis (LCMS 8060, Shimadzu Italy, Milan, Italy). The MS/MS used electrospray ionization and was controlled by Lab Solution software, which allowed for rapid change from a low-energy scan at 4 V (full scan MS) to a high-energy scan (10–60 V ramping) during a single LC run. The following source parameters were set: 2.9 L/min nebulizing gas flow, 10 L/min heating gas flow, 300 °C interface temperature, 250 °C DL temperature, 400 °C heat block temperature, and 10 L/min drying gas flow. The analysis was carried out by flow injection (i.e., no chromatographic separation), with acetonitrile: water + 0.01 percent formic acid (5:95, *v/v*) as the mobile phase. Instrument was set to positive mode for an SIM experiment [48,49]. Samples were considered “positive” if the area under the curve was higher in magnitude than the blank. Pure standards used are listed in Appendix A. Ergovaline, galanthamine, nor-galanthamine, anhydrolycorine, trisphaeridine, crinine, demethylmaritidine, N-formylgalanthamine and peramine were purchased from Sigma Aldrich.

### 4.3. Pharmacokinetic Parameters and Toxicity Prediction

Pharmacokinetic properties and toxicity of alkaloids identified in the hydroethanolic extract of *D. ambrosioides* flowers using LC-MS/MS were predicted using the Swiss ADME (http://www.swissadme.ch/) (accessed on 1 May 2022) and pkCSM (http://biosig.unimelb.edu.au/pkcsm/prediction) (accessed on 1 May 2022) online tools [50].

### 4.4. Experimental Animals

Wistar rats and albino mice were raised atthe Department of Biology (Faculty of Sciences, Oujda, Morocco) and divided into experimental groups with a sex ratio of 1 (♂/♀ = 1), then placed under standard conditions of constant temperature (22 ± 2 °C), with 12 h of light and 12 h of darkness and free access to water and food. The animal experiments were carried out in accordance with the United States National Institutes of Health’s internationally recognized guide for the care and use of laboratory animals (NIH Publication No. 85–23, revised 1985). The study was carried out in accordance with the Declaration of Helsinki’s criteria and was authorized by the Faculty of Sciences’ institutional review board in Oujda, Morocco (01/20-LBBEH-04 and 9 January 2020).

### 4.5. Acute Toxicity Studies in Mice

The OECD guidelines were used to evaluate the single-dose acute oral toxicity study (425) [46]. Male and female albino mice (20–35 g) were used in the acute toxicity investigation and were randomly divided into three groups of six mice each; all mice in each group were administered a single oral dose. Each group received a single oral dose of DAHE of 1, 2, 3, 5, or 7 g/kg body weight. The control group was given 1 mL of distilled water per 100 g of animal weight. After a single oral administration of DAHE, the mice were returned to the cages and observed every 30 min for 4 h and then once daily for 15 days, observing behavioral changes, mortality, and/or symptomatic disorders, such as restlessness, anorexia, motor difficulties, skin appearance, hair, weight, etc. After this period of observation, the mice of all groups were anesthetized and dissected to determine the general appearance of the organs and to note their weight. Furthermore, the LD_50_ was calculated using the Dragstedt and Lang method, as reported in [51].

### 4.6. Subchronic Toxicity

#### 4.6.1. Treatments

For the study of sub-chronic toxicity, Wistar rats were divided into four groups of six rats each: control; oral treatment with distilled water for 15 days; and test, treated by gavage for 15 days with the following doses of DAHE: 5, 50, and 500 mg/kg body weight. The goal of choosing these levels was to establish LD_50_, in addition to calculating the dose suggested in by OECD Guideline 407 [52].The animals were evaluated daily for general health manifestations and clinical toxicity symptoms over during the15-day study period, and body weight changes were recorded on days 0, 7, 14, 21, and 28. All animals were fasted overnight after 15 days of treatment, and blood samples were obtained from the abdominal artery in anticoagulant tubes while anesthetized with ether. The serum was also separated for biochemical analysis by centrifuging the tubes for 10 min at 3000 rpm. The organs (heart, liver, kidney, and lung) were also elevated for histological investigation. The relative weights of organs (liver, kidneys, heart, and lungs) were also measured.

#### 4.6.2. Serum Biochemistry

Clinical diagnostic kits and manufacturer methods were used for biochemical analysis of serum samples. The measured parameters were albumin (ALB), alkaline phosphatase (ALP), alanine aminotransferase (ALT), aspartate transaminase (AST), bilirubin (BIL), cholesterol (CHOL), triglycerides (TRGL), creatinine (CRE), and urea (URE).

#### 4.6.3. Histopathological Examination of the Organs

Following sacrifice, the liver, kidneys, heart, and lungs were removed immediately, weighed (absolute organ weight), fixed in formalin 10% for 3±1 days, embedded in paraffin wax, sectioned into 3–4 µm sections, and stained with hematoxylin and eosin. The colored sections were observed using optical microscopy (Optika Microscopes, Italy) and captured by an Infinity 1 camera microscope under objective 40. The relative organ weight (ROW) of each animal was calculated as follows [53]:(1)$$\mathrm{Relative}\;\mathrm{Organ}\;\mathrm{Weight}\;\left(\%\right)=\frac{\mathrm{Organ}\;\mathrm{weight}\;\left(g\right)}{\mathrm{Body}\;\mathrm{weight}\;\left(g\right)}\times 100$$

Histopathological study includes observing tissue integrity and looking for injuries, such as degeneration, necrosis, apoptosis, and leukocyte infiltration, which could be a symptom of toxicity.

### 4.7. Statistical Analysis

The mean and standard error of the mean were used to describe the data (SEM). Analysis of variance (one-way ANOVA) and Dunnett’s test were used to identify differences between groups in the subacute toxicity test. The level of significance was fixed at 0.05. GraphPad Prism software version 8.4.3 for Windows was used to conduct the statistical analysis.

## 5. Conclusions

We examined the acute and subacute toxicity of DAHE after oral treatment in mice and rats. DAHE had an oral LD_50_ of 5000 mg/kg. A 15-day oral subacute toxicity investigation in rats revealed that *D. ambrosioides* hydroethanolic extract is relatively safe when administered orally. According to the LC-MS/MS study, this extract includes nine alkaloids. However, isolation of these alkaloids should be performed to determine the toxicity and safety of this plant.

Finally, it is critical to understand that medicinal plants should be studied and evaluated for toxicity and safety. These findings provide valuable preliminary information on the toxicological profile of *D. ambrosioides*. As a result, more testing (such as hematological parameters, genotoxicity, subchronic toxicity, reproductive toxicity, and component toxicity) is needed before moving forward, with clinical trials of this plant.

## Figures and Tables

**Figure 1 toxins-14-00475-f001:**
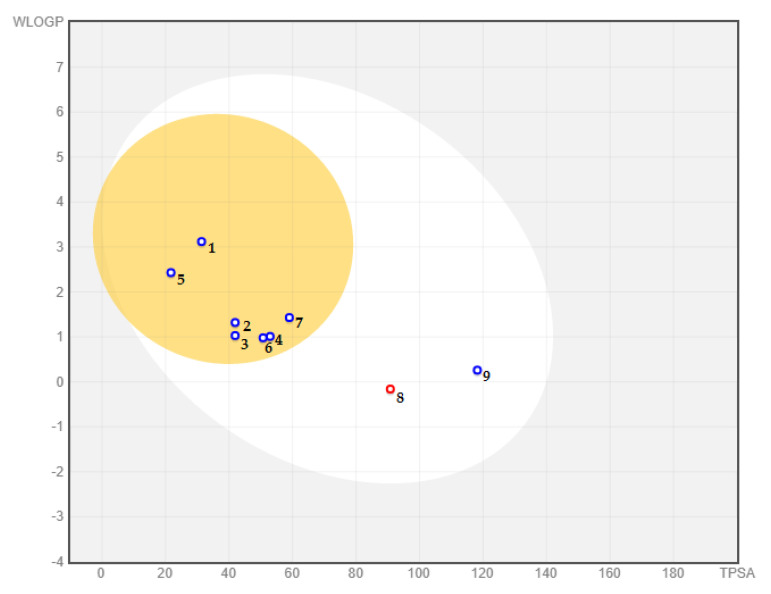
BOILED-Egg predicting model for blood–brain barrier permeability and intestinal absorption of the molecules: (**1**) trisphaeridine, (**2**) galanthamine, (**3**) crinine, (**4**) demethylmaritidine, (**5**) anhydrolycorine, (**6**) nor-galanthamine, (**7**) N-formylnorgalanthamine, (**8**) peramine, and (**9**) ergovaline.

**Figure 2 toxins-14-00475-f002:**
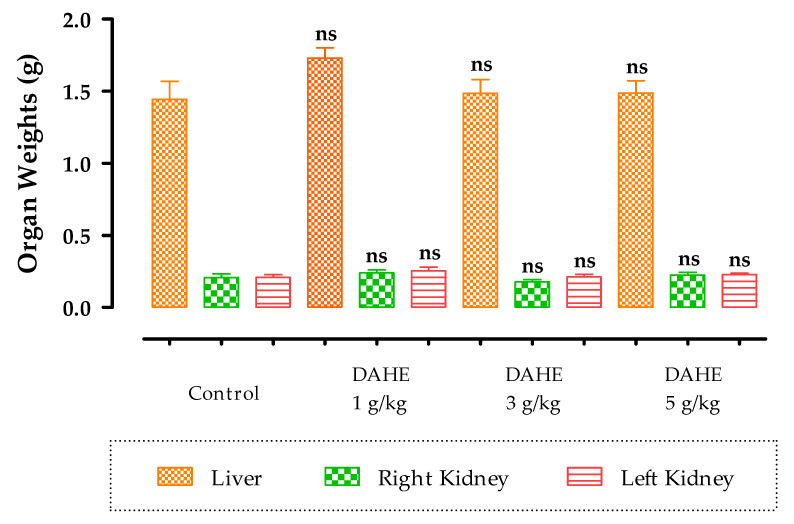
The effect of DAHE on the relative weight of organs in mice. The data are shown as mean ± standard error of the mean (six animals per group). ns: non-significant; no significant difference was found when comparing treated groups to the control group.

**Figure 3 toxins-14-00475-f003:**
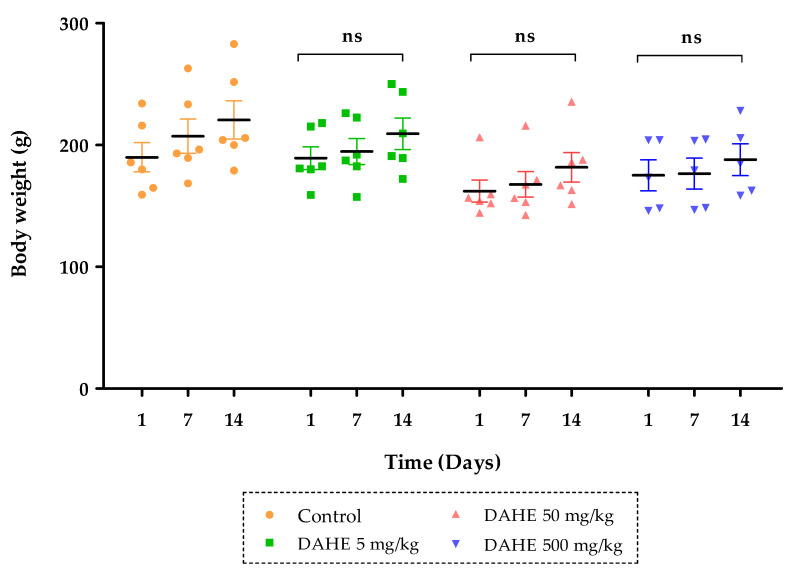
Changes in body weight in rats treated with DAHE during the 15-day subacute toxicity study. The results are presented as the mean ± SEM (n = 6). ns: non-significant.

**Figure 4 toxins-14-00475-f004:**
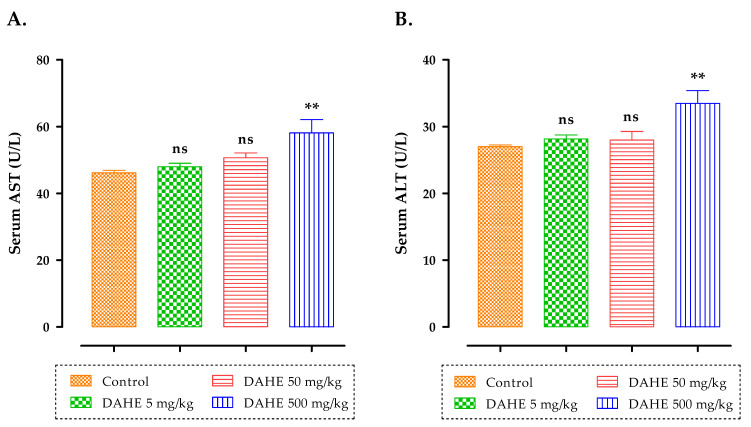
Effect of subacute oral administration of DAHE on (**A**) AST (U/L) and (**B**) ALT in serum. Values are presented as the means ± SEM. When compared to the control group, there were significant differences. ns: non-significant, ∗∗ *p* < 0.01.

**Figure 5 toxins-14-00475-f005:**
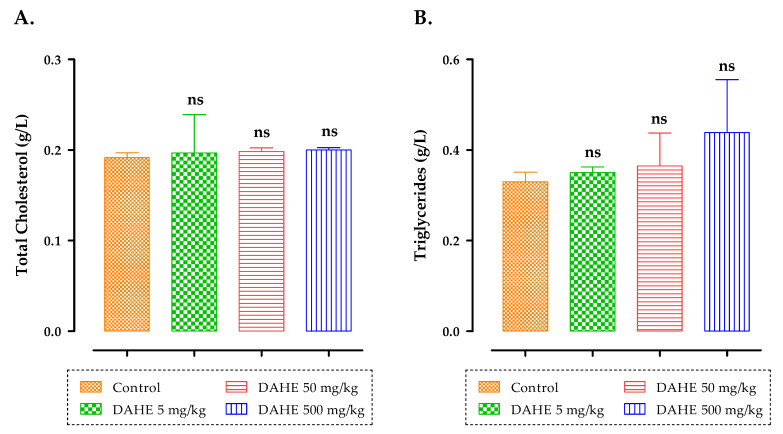
Total cholesterol (**A**) and triglycerides (**B**) after a subacute oral dose of DAHE. Data are presented as means and standard deviations (SEM). When compared to the control group, there were no significant differences. ns: non-significant.

**Figure 6 toxins-14-00475-f006:**
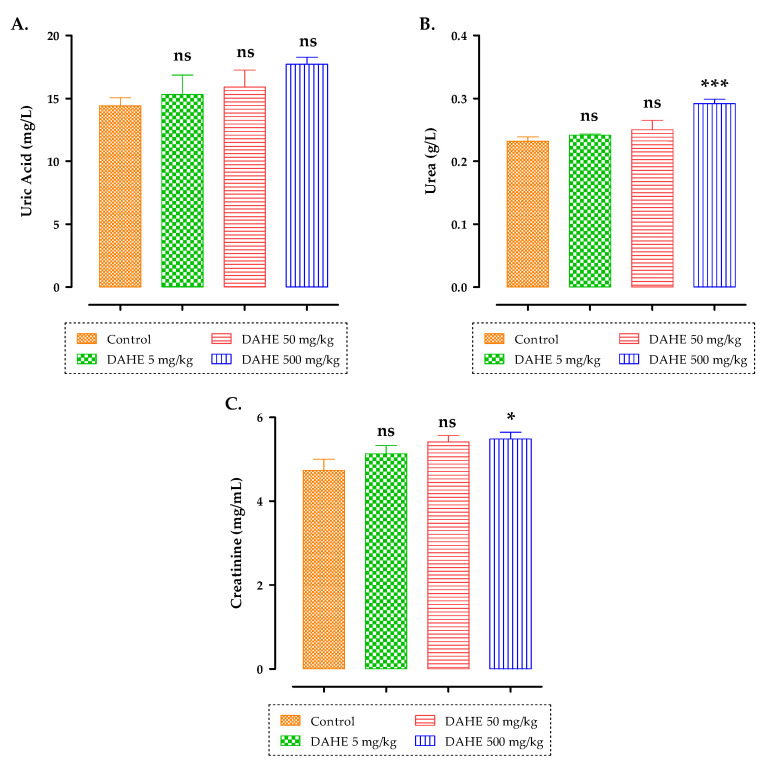
The effect of oral administration of DAHE on uric acid (**A**), urea (**B**), and creatinine (**C**) in treated rats. Data are presented as mean ± SEM. When compared to the control group, there were significant differences. ns: non-significant, * *p* < 0.05 and *** *p* < 0.001.

**Figure 7 toxins-14-00475-f007:**
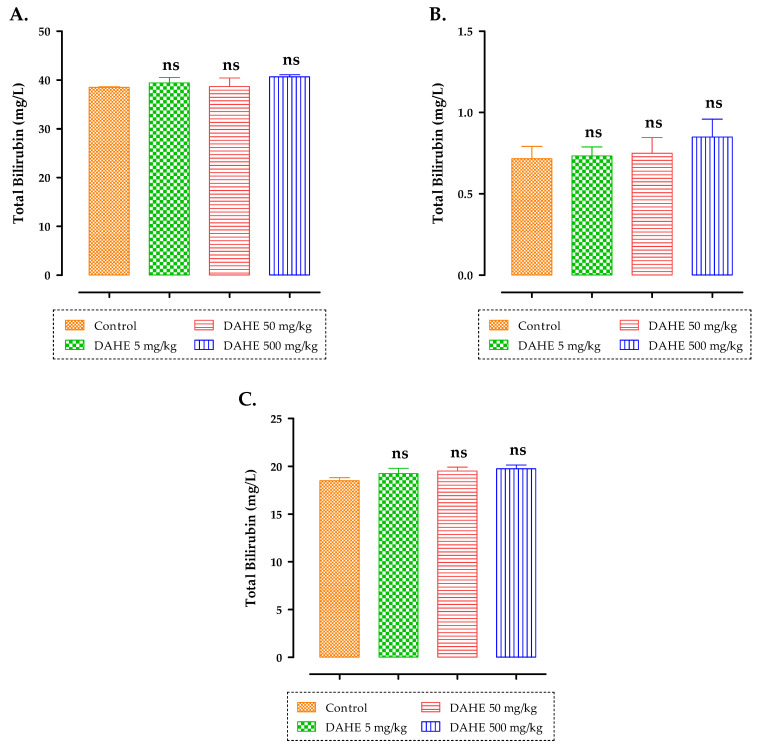
Effect of subacute oral administration of DAHE on total protein (**A**), total bilirubin (**B**), and albumin (**C**) in rats. ns: non-significant.

**Figure 8 toxins-14-00475-f008:**
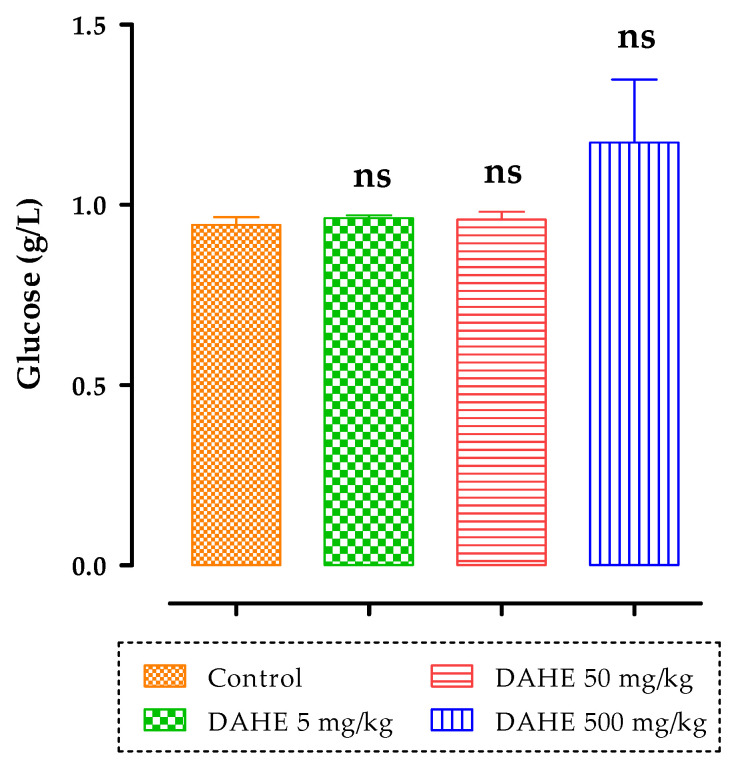
Effect of subacute oral administration of DAHE on plasma glucose levels in rats. ns: non-significant.

**Figure 9 toxins-14-00475-f009:**
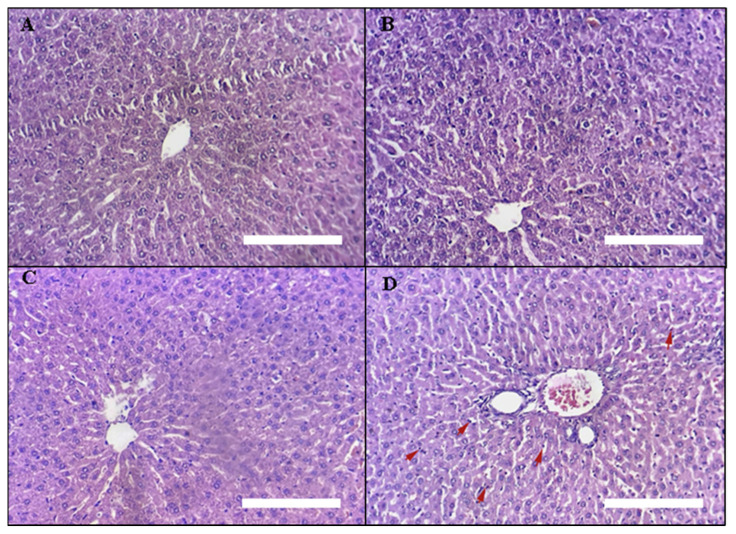
Liver histopathology. Light micrographs of liver sections of different treatment groups. The treatment groups are represented by the numbers on the photos. (**A**) untreated rats, (**B**) rats treated with5 mg/kg DAHE, (**C**) rats treated with 50 mg/kg DAHE, and (**D**) rats treated with 500 mg/kg DAHE. Red arrows = inflammatory cells (granulocytes and lymphocytes).White scale bar = 1.03 mm.

**Figure 10 toxins-14-00475-f010:**
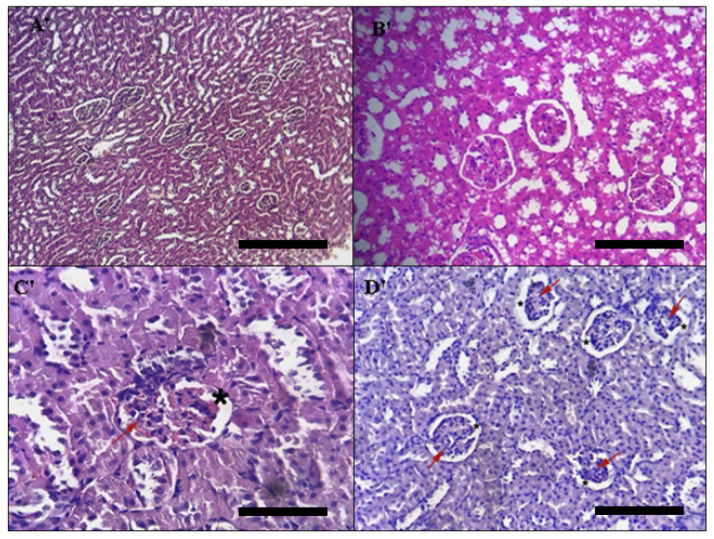
Histopathology of the kidney. Light micrographs of liver slices of different treatment groups. The treatment groups are represented by the numbers on the photos. (**A′**) untreated rats, (**B′**) rats treated with 5 mg/kg DAHE, (**C′**) rats treated with 50 mg/kg DAHE, and (**D′**) rats treated with 500 mg/kg DAHE. Red arrow = decrease in glomerulus cells; Black star = expansion of the Bowman space. Black scale bar = 1.03 mm.

**Figure 11 toxins-14-00475-f011:**
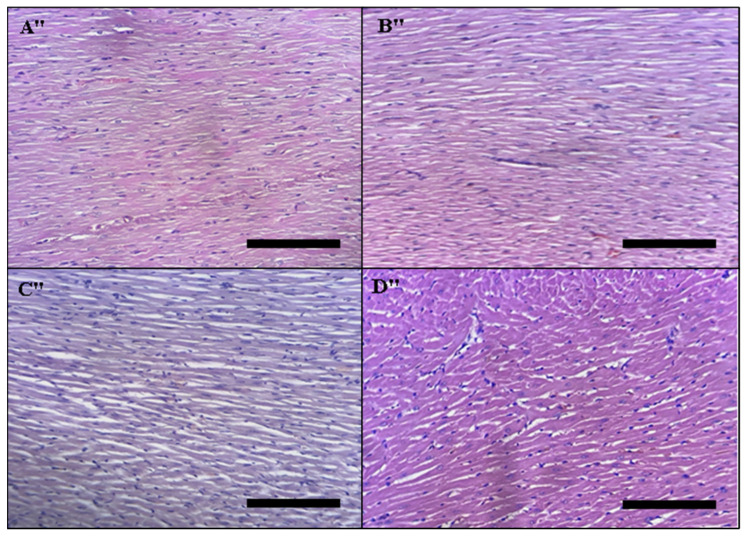
Histopathology of the heart. Light micrographs of heart slices of different treatment groups. The treatment groups are represented by the numbers on the photos. (**A****″**) Untreated rats, (**B****″**) rats given the DAHE (5 mg/kg), (**C****″**) rats treated with the DAHE (50 mg/kg), and (**D****″**) rats treated with the DAHE (500 mg/kg).Black scale bar = 1.03 mm.

**Figure 12 toxins-14-00475-f012:**
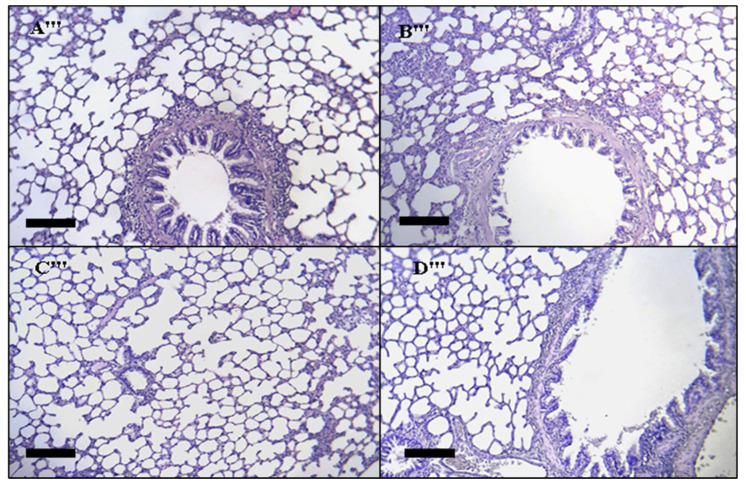
Histopathology of the lung. Lung slice light micrographs of different treatment groups. The treatment groups are denoted by numbers on the photographs. (**A****‴**), rats treated with 5 mg/kg DAHE, (**B****‴**) control rats, (**C****‴**) rats treated with 50 mg/kg DAHE, and (**D****‴**) rats treated with 500 mg/kg DAHE. Black scale bar = 0.51 mm.

**Table 1 toxins-14-00475-t001:** Alkaloid composition of LC-MS/MS *D. ambrosioides* hydroethanolic extract.

N°	Molecules	Molecular Formula	Selected [M-H]^+^	Literature [M-H]^+^	RT (min)	Abundance
**1**	Trisphaeridine	C_14_H_9_NO_2_	223.000	223.000 [27]	0.338	++
**2**	Galanthamine	C_17_H_21_NO_3_	286.000	288.159 [27] 287.000 [28]	0.320	++
**3**	Crinine	C_16_H_17_NO_3_	271.000	271.000 [27]	0.327	++
**4**	Demethylmaritidine	C_16_H_19_NO_3_	273.000	273.000 [27] 272.900 [29]	0.323	++
**5**	Anhydrolycorine	C_16_H_13_NO_2_	250.000	251.000 [27]	0.337	++
**6**	Nor-galanthamine	C_16_H_19_NO_3_	272.000	274.143	0.326	++
**7**	N-formylnorgalanthamine	C_17_H_19_NO_3_	301.000	301.000 [27]	0.321	+++
**8**	Peramine	C_12_H_17_N_5_O	248.200	248.150 [30]	0.336	+
**9**	Ergovaline	C_29_H_35_N_5_O_5_	543.300	534.000 [31]	0.323	+

+++: High abundance, ++: abundant, +: low abundance.

**Table 2 toxins-14-00475-t002:** Absorption, distribution, metabolism, excretion, and toxicity prediction of alkaloids from *D. ambrosioides* hydroethanolic extract: (**1**) trisphaeridine, (**2**) galanthamine, (**3**) crinine, (**4**) demethylmaritidine, (**5**) anhydrolycorine, (**6**) nor-galanthamine,(**7**) n-formylnorgalanthamine, (**8**) peramine, and (**9**) ergovaline.

Compound No.		1	2	3	4	5	6	7	8	9
**DrugLikeness**	**Lipinski’s Rule of Five**	Yes	Yes	Yes	Yes	Yes	Yes	Yes	Yes	Yes
	**Bioavailability Score (%)**	0.55	0.55	0.55	0.55	0.55	0.55	0.55	0.55	0.55
**Absorption**	**Water Solubility**	−3.91	−2.93	−2.82	−2.64	−3.81	−2.55	−3.17	−1.11	−4.59
	**Caco2 Permeability**	1.09	1.59	1.81	1.16	1.76	1.17	1.24	0.75	0.11
	**Intestinal Absorption (Human) (%)**	98.97	94.99	93.80	94.58	98.05	94.77	97	77.02	68.16
	**Skin Permeability**	−5.33	−6.75	−6.74	−6.90	−5.64	−6.99	−6.92	−8.23	−7.83
	**P-glycoprotein Substrate**	Yes	Yes	Yes	Yes	Yes	Yes	Yes	No	Yes
	**P-glycoprotein Inhibitor**	No	No	No	No	No	No	No	No	Yes
	**P-glycoprotein Inhibitor**	No	No	No	No	No	No	No	No	No
**Distribution**	**VDss(human)**	−0.02	0.89	0.88	1.05	0.66	0.88	0.14	0.55	0.91
	**BBBpermeability**	0.15	−0.08	−0.02	−0.15	0.25	−0.09	−0.32	−0.75	−0.64
	**CNS permeability**	−1.56	−2.51	−2.47	−2.92	−1.57	−2.90	−2.93	−3.36	−2.89
**Metabolism**	**CYP2D6 Substrate**	No	No	Yes	Yes	No	No	No	No	No
	**CYP3A4 Substrate**	Yes	Yes	Yes	Yes	Yes	Yes	Yes	No	Yes
	**CYP2D6 Inhibitor**	Yes	Yes	Yes	Yes	No	Yes	Yes	No	Yes
	**CYP3A4 Inhibitor**	Yes	No	No	No	No	No	No	No	No
**Excretion**	**Total Clearance**	0.21	0.99	1.12	1.07	0.14	1.09	1.02	0.58	0.47
	**Renal OCT2 Substrate**	No	Yes	No	No	No	No	No	No	No
**Toxicity**	**AMES Toxicity**	Yes	No	Yes	No	Yes	No	No	Yes	No
	**Hepatotoxicity**	Yes	Yes	Yes	No	No	Yes	No	No	No
	**hERG I Inhibitor**	No	No	No	No	No	No	No	No	No
	**hERG II inhibitor**	No	Yes	No	No	No	No	No	No	Yes
	**Skin Sensitization**	No	No	No	No	No	No	No	No	No
	**Maximal Tolerated Dose**	0.29	−0.42	−0.56	−0.77	−0.37	−0.33	−0.49	−0.16	−0.84
	**Oral Rat Acute Toxicity (LD50)**	2.61	2.72	2.47	2.75	2.25	2.64	3.01	2.54	3.17
	**Oral Rat Chronic Toxicity (LOAEL)**	2.03	0.96	1.2	1.45	1.58	1.25	0.69	2.31	2.81
	**Minnow Toxicity**	−0.32	1.67	2.24	2.09	0.42	2.09	2.08	1.89	3.37
	***T. pyriformis* toxicity**	0.35	0.78	0.76	0.28	1.24	0.39	0.45	0.27	0.28

**Table 3 toxins-14-00475-t003:** Acute toxicity, mortality, and clinical signs in mice following oral administration of *D. ambrosioides* hydroethanolic extract.

DAHE Dose (g/kg)	Mortality	Signs of Toxicity
Control (none)	0/6	Normal
1	0/6	Normal
2	0/6	Normal
3	1/6	Hypoactivity/anorexia
5	3/6	Hypoactivity/anorexia
7	6/6	Death

**Table 4 toxins-14-00475-t004:** Alterations in urinary volume and food and water intake in rats treated with DAHE during the 15-day subacute toxicity study. Results are presented as mean ± SEM with n = 6. ns: non-significant.

	Control	DAHE (5 mg/kg)	DAHE (50 mg/kg)	DAHE (500 mg/kg)
**Food intake (g)**	60.35 ± 3.58	57.22 ± 5.32 ^ns^	48.63 ± 4.86 ^ns^	47.47 ± 3.01 ^ns^
**Water intake (mL)**	30.00 ± 3.65	31.67 ± 4.01 ^ns^	34.33 ± 6.45 ^ns^	43.33 ± 4.94 ^ns^
**Urinary volume (mL)**	5.20 ± 0.37	5.66 ± 0.84 ^ns^	6.16 ± 1.24 ^ns^	6.33 ± 0.71 ^ns^

**Table 5 toxins-14-00475-t005:** Absolute and relative organ weights of rats treated orally with DAHE. ns = non-significant.

Parameter		Control	DAHE (5 mg/kg)	DAHE (50 mg/kg)	DAHE (500 mg/kg)
**Absolute weight of organs**	**Liver (g)**	7.34 ± 0.46	7.41 ± 0.85 ^ns^	7.10 ± 0.99 ^ns^	6.72 ± 1.30 ^ns^
	**Kidney (g)**	0.60 ± 0.07	0.62 ± 0.10 ^ns^	0.56 ± 0.12 ^ns^	0.55 ± 0.12 ^ns^
	**Heart (g)**	0.81 ± 0.10	0.75 ± 0.09 ^ns^	0.69 ± 0.07 ^ns^	0.71 ± 0.16 ^ns^
	**Lung (g)**	1.23 ± 0.20	1.22 ± 0.09 ^ns^	1.21 ± 0.24 ^ns^	1.11 ± 0.13 ^ns^
**Relative weight of organs**	**Body weight (g)**	220.6 ± 5.06	207.16 ± 2.04 ^ns^	188.33 ± 3.03 ^ns^	181.88 ± 1.09 ^ns^
	**Liver (g)**	3.33 ± 0.18	3.58 ± 0.41 ^ns^	3.77 ± 0.53 ^ns^	3.70 ± 0.71 ^ns^
	**Kidney (g)**	0.27 ± 0.03	0.30 ± 0.05 ^ns^	0.29 ± 0.06 ^ns^	0.30 ± 0.06 ^ns^
	**Heart (g)**	0.37 ± 0.04	0.36 ± 0.04 ^ns^	0.36 ± 0.04 ^ns^	0.39 ± 0.09 ^ns^
	**Lung (g)**	0.56 ± 0.09	0.59 ± 0.05 ^ns^	0.64 ± 0.13 ^ns^	0.61 ± 0.07 ^ns^

## Data Availability

All data supporting the findings of this study are included in this article.

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
