# Peer review of "Evaluation of Acute and Subacute Toxicity and LC-MS/MS Compositional Alkaloid Determination of the Hydroethanolic Extract of Dysphania ambrosioides (L.) Mosyakin and Clemants Flowers"

_toxins, 2022, doi:10.3390/toxins14070475_

Round 1

Reviewer 1 Report

The present manuscript dealt with evaluation of acute and subacute toxicity and LC-MS/MS compositional alkaloids determination of the hydroethanolic extract of Dysphania ambrosioides(L.) Mosyakin and Clemants flowers. The reviewer read carefully and raised some concerns as below:

  1. In Materials and Methods section, the authors should deposit their sample as a voucher specimen at an authentic institution for reference.
  2. In animal exp., authors should provide their permission number of Animal Ethics Comittee. 
  3. In Fig.s 3~9, why did the authors display the negative data? Should be deleted.
  4. In Figs. 10~13, please show the exact molecular toxic mechanisms derived from the extracts. Only phenomenal presentation is not acceptable for publication. 
  5. Resubmission is strongly recommended after tentative revision. 

Author Response

Reviewer 1: The present manuscript dealt with the evaluation of acute and subacute toxicity and LC-MS/MS compositional alkaloids determination of the hydroethanolic extract of Dysphania ambrosioides (L.) Mosyakin and Clemants flowers. The reviewer read carefully and raised some concerns as below:

  1. In Materials and Methods section, the authors should deposit their sample as a voucher specimen at an authentic institution for reference.
  2. In animal exp., authors should provide their permission number of Animal Ethics Committee.
  3. In Figs 3~9, why did the authors display the negative data? Should be deleted.
  4. In Figs. 10~13, please show the exact molecular toxic mechanisms derived from the extracts. Only a phenomenal presentation is not acceptable for publication.

Resubmission is strongly recommended after tentative revision.

Authors: We highly appreciate your positive feedback on our manuscript. we believe that your remarks have helped to improve the quality of our manuscript so it can be suitable for publication in Toxins. We have paid particular attention to all your comments and suggestions, and provided you an answer to each below as well as in the revised version of the manuscript appearing in yellow.

  1. We have deposited voucher specimens in the herbarium of the faculty of sciences under the reference HUMPOM44).
  2. We thank the reviewer for this remark; thus, we have added to the manuscript the permission number of the Animal Ethics Committee (Lines 406-408).
  3. Negative data in this paper are also valuable results; they have translated the fact that there is no toxicity in the hydroethanolic extract of D. ambrosioides’ flowers.
  4. Thank you for pointing this out, we have added that the changes remarked in rats’ tested organs are in line with the biochemical changes noticed. Extra parts of the text were added to the manuscript and highlighted in yellow.

Reviewer 2 Report

The authors report an interesting phytochemical and toxicological study on hydroalcoholic extracts of Dysphania ambrosioides.

Initially the alkaloids present in the hydroalcoholic extract were evaluated in LC-MS-MS, subsequently an ADME in silico study was performed. Finally, toxicity was assessed acute in mice and chronic in rats.

The study is interesting and the applied methodologies are correct. The results obtained are clear. However, some methodological details need to be clarified.

In particular:

1) the authors evaluated the body weight and food consumption of rats after repeated administration of the hydroalcoholic extract. However, data on daily changes in food and water consumption have not been reported. This parameter is essential when performing oral administrations.

2) Authors should enter more details on the animals used, for example: Genetic strain and sex. Why did they use males and females? Did they also use female rats in the chronic treatment?

3) In the introduction it is necessary to add further references on the use of hydro-alcoholic extracts. In this regard the authors should add: 

Abate, G. et al.. Phytochemical Analysis and Anti-Inflammatory Activity of Different Ethanolic Phyto-Extracts of Artemisia annua L.. Biomolecules 202111, 975. doi: 10.3390/biom11070975

Author Response

Reviewer 2: The authors report an interesting phytochemical and toxicological study on hydroalcoholic extracts of Dysphania ambrosioides. Initially, the alkaloids present in the hydroalcoholic extract were evaluated in LC-MS/MS, subsequently, an ADME in silico study was performed. Finally, toxicity was assessed acute in mice and chronic in rats.

The study is interesting and the applied methodologies are correct. The results obtained are clear. However, some methodological details need to be clarified.

  1. the authors evaluated the body weight and food consumption of rats after repeated administration of the hydroalcoholic extract. However, data on daily changes in food and water consumption have not been reported. This parameter is essential when performing oral administrations.
  2. Authors should enter more details on the animals used, for example, Genetic strain and sex. Why did they use males and females? Did they also use female rats in the chronic treatment?
  3. In the introduction it is necessary to add further references on the use of hydro-alcoholic extracts. In this regard the authors should add:

Abate, G. et al. Phytochemical Analysis and Anti-Inflammatory Activity of Different Ethanolic Phyto-Extracts of Artemisia annua L. Biomolecules 2021, 11, 975. doi: 10.3390/biom11070975

Authors: Dear Reviewer 2, we highly appreciate your positive feedback on our work, we believe that the manuscript is well improved after addressing your comments. we have revised the manuscript accordingly. All the raised specific observations were toked into consideration and were highlighted in yellow.

  1. We agree with your observation, but we evaluated food and water consumption after treatment with metabolic cages, and a comparison was made between the test groups and the control treated with distilled water. and the results we obtained provide a general idea about the daily estimation of these parameters, the results revealed that there is no significant difference see table 4.
  2. We appreciate your comment; the animal details have been moved to paragraph 4.4; we have worked on males and females to evaluate the toxicity by diversifying the sex because the physiology between the two differs; additionally, it is about toxicity on the animal model, not toxicity targeting the sexes, for which we have worked 50% male and 50% female; this allows us to have a general idea on the toxicity of our extract on the sexes.
  3. We have added to the introduction the necessary information needed. We highly appreciate the reviewer’s remark.

Reviewer 3 Report

Biological part of experiments is interesting; however, the main drawback is poor characteristic of extract composition. Authors have a powerful tool such as LC-MS; so the results should be more detailed. Hydroethanolic extract probably contained also polyphenols.

 Detailed suggestions:

1)      On what basis the identification of alkaloids was performed? Mass with accuracy to two decimal places is not enough to generate formula. Moreover (Table 1), two alkaloids have the same molecular mass.

2)      Line 53: „As a consequence of phytochemical screening, the genus Chenopodium has produced substances…” – not precise statement. Phytochemical screening is responsible for production of metabolites? Reedit

3)      Line 70: „chromatographic analysis” – this is not chromatographic analysis because chromatography is based on separation between stationary and mobile phase; meanwhile in the study there was no separation

4)      Line 71: “in large quantities” – no quantitative analysis was performed. On what basis such statement? (the same comment for “in small quantities” )

5)      Table 1: RT is unnecessary because no separation was performed. On what basis abundance was assessed?

6)      Fig.3. It should be clearly indicated in figure legend that the differences was insignificant

7)      Fig. 4: explain the abbreviation “ns” under the Table (the same for Figs. 5-8)

8)      Line 390: “ in negative mode (only syringic acid in positive ESI)” – alkaloids are usually analyzed in positive mode. Why syringic acid was mention?

9)      How was prepared extract for biological assay? What solvent was used to dissolved dry residue obtained in 4.1 section? What was proportion of solvent to solid?

10)   Line 412: what does it mean “1 cc”?

11)   It is not clear what is shown in Supplementary material. This is not chromatogram. M/z scan? Add some details.

12)   There are some editorial errors, e.g. lack of spaces (e.g. lines 34,36,37,150, Table 5.…) line 63: „brfore”, unnecessary dot (line 157, 377) and unnecessary capital letter (line 164), lack of italic (395, 467)

Author Response

Dear Editors, Dear Reviewers,

Thank you for giving us the opportunity to improve our manuscript with the revised version and thank you for your useful comments.

We really appreciate the Reviewers’ comments. We hope this revision will satisfy reviewers’ queries, and that our work will be considered for publication in Toxins.

With kind regards

Reviewer 3: Biological part of the experiments is interesting; however, the main drawback is poor characteristics of extract composition. Authors have a powerful tool such as LC-MS; so the results should be more detailed. Hydroethanolic extract probably contained also polyphenols.

 Detailed suggestions:

  1. On what basis the identification of alkaloids was performed? Mass with accuracy to two decimal places is not enough to generate a formula. Moreover (Table 1), two alkaloids have the same molecular mass
  2. Line 53: „As a consequence of phytochemical screening, the genus Chenopodium has produced substances…” – not a precise statement. Phytochemical screening is responsible for the production of metabolites? Reedit
  3. Line 70: „chromatographic analysis” – this is not chromatographic analysis because chromatography is based on separation between stationary and mobile phase; meanwhile in the study there was no separation
  4. Line 71: “in large quantities” – no quantitative analysis was performed. On what basis such statement? (the same comment for “in small quantities” )
  5. Table 1: RT is unnecessary because no separation was performed. On what basis abundance was assessed?
  6. 3. It should be clearly indicated in figure legend that the differences was insignificant
  7. 4: explain the abbreviation “ns” under the Table (the same for Figs. 5-8)
  8. Line 390: “ in negative mode (only syringic acid in positive ESI)” – alkaloids are usually analyzed in positive mode. Why syringic acid was mention
  9. How was prepared extract for biological assay? What solvent was used to dissolved dry residue obtained in 4.1 section? What was proportion of solvent to solid?
  10. Line 412: what does it mean “1 cc”?
  11. It is not clear what is shown in Supplementary material. This is not chromatogram. M/z scan? Add some details.
  12. There are some editorial errors, e.g. lack of spaces (e.g. lines 34,36,37,150, Table 5.…) line 63: „brfore”, unnecessary dot (line 157, 377) and unnecessary capital letter (line 164), lack of italic (395, 467)

Authors: Thank you so much for your positive comments on our manuscript, we hope our answers have clarified our point and improved the quality of this work, in order to make it suitable for publication in Applied Sciences.

  1. The identification of compounds was carried out by analyzing their molecular weight in the [M-H]- form (negative ion mode) as extensively described in the literature. There is a table with the transition of each alkaloid and the related reference). Raw data in the supplemental material explain the result of table 1 due to the direct correlation of the area under the curve (elicited in the upper portion of the graph) with the abundance of compounds.
  2. We thank reviewer 3 for his remark. We have reedited it in the manuscript.
  3. This error was well corrected in the manuscript.
  4. Thank you for your thoughtful comments, dear reviewer. We agree with your comment, and after doing qualitative phytochemical analysis, we have determined that it is preferable to avoid using little and large quantities, thus we will make the necessary changes.
  5. The relative abundance of each alkaloid was estimated by analyzing peak area (AUC –area under the curve). The intensity of the signal relates to compound abundance (concentration).
  6. Thank you for your comment, we have updated figure 3. And we have added to the title the meaning of what we have added (ns: non-significant).
  7. We appreciate your remark, and we have mentioned the meaning of “ns” under each figure.
  8. We have made the necessary changes, thank you.
  9. For all of the experiments, we used the gavage rule of 1ml per 100g of the animal's body weight. We dissolved the residue in distilled water just because the extract is soluble in water, avoiding the installation of an intoxicant by the solvent.
  10. This error was fixed accordingly, we meant 1 mL.
  11. We have added some details to the supplementary material, thank you for pointing this out.
  12. Thank you for your comment, we have fixed the editorial errors raised.

Round 2

Reviewer 1 Report

Thank you for your revision. Please revise the followings as below;

1. In Figs. 10~13, could not find scale bars in the images. 

In the each legend of Figs. 10~13 , "500 mg/kg" is correct. 

2. There are so many typos in the text. For example, In Conclusion section, Line 488, "it's" should be changed to "it is".  

Author Response

Dear Editors, Dear Reviewers,

Thank you for allowing us a second opportunity to improve our manuscript.We have reviewed our manuscript and made all required changes as inthe Reviewers’ comments. We hope this revision will satisfy reviewers’ queries, and that our work will be considered for publication in Toxins.

With kind regards

Dr.Hano, Dr.Addi, and the co-Authors.

Reviewer 1:Thank you for your revision. Please revise the followings as below;

  1. In Figs. 10~13, could not find scale bars in the images. In each legend of Figs. 10~13, "500 mg/kg" is correct.
  2. There are so many typos in the text. For example, In Conclusion section, Line 488, "it's" should be changed to "it is".

Authors: Dear Reviewer 1, thank you so much for your positive feedback on our manuscript. We have paid particular attention to all your comments and suggestions. We have added to Figs 10-13, scale bars, and we have revised the manuscript accordingly to correct all typos in it.

Reviewer 3 Report

Manuscript has been improved but still analytical part raises doubts.

Lack of information (references) on what basis the identification was done. In Authors response is stated: “The identification of compounds was carried out by analyzing their molecular weight in the [M-H]- form (negative ion mode) as extensively described in the literature” – however no references is included. The ionization mode in MS is also unusual - alkaloids more easily accept protons and therefore, they are commonly analyzed in a positive mode.

The masses in Table 1 is also incorrect. Exact mass (not molecular weight) should be given as a result of MS analysis.

Supplementary is still incorrect. This is not chromatographic analysis because no chromatography was performer. Moreover, m/z given for particular compounds differ from values in Table (see e.g. last compound: in Table is Ergovaline m/z 533 and in Supp. Material is m/z 543

Minor:

Line 158: Unnecessary dot

Line 228: lack of space „(500mg/kg)”

Author Response

Dear Editors, Dear Reviewers,

Thank you for allowing us a second opportunity to improve our manuscript. We have reviewed our manuscript and made all required changes as in the Reviewers’ comments. We hope this revision will satisfy reviewers’ queries, and that our work will be considered for publication in Toxins.

With kind regards

Reviewer 3: Manuscript has been improved but still analytical part raises doubts.

Lack of information (references) on what basis the identification was done. In Authors response is stated: “The identification of compounds was carried out by analyzing their molecular weight in the [M-H]- form (negative ion mode) as extensively described in the literature” – however no references is included. The ionization mode in MS is also unusual - alkaloids more easily accept protons and therefore, they are commonly analyzed in a positive mode.

  1. The masses in Table 1 is also incorrect. Exact mass (not molecular weight) should be given as a result of MS analysis.
  2. Supplementary is still incorrect. This is not chromatographic analysis because no chromatography was performer. Moreover, m/z given for particular compounds differ from values in Table (see e.g. last compound: in Table is Ergovaline m/z 533 and in Supp. Material is m/z 543

Minor:

  1. Line 158: Unnecessary dot
  2. Line 228: lack of space „(500mg/kg)”

Authors: Dear Reviewer 2, we highly appreciate your comments on our paper, and we thank you for pointing errors in our manuscript.

  1. As stated in the spectra enclosed to the supplementary material, the utilized ionization mode was ESI+. The text had a typo. The paragraph was corrected according to the suggestion. Thank you so much.
  2. We really appreciate your remarks and we accept the revision. Table 1 was corrected as well according to the given indications. In particular, the selected [M-H]+ derive from a process of optimization and selection controlled by the mass chromatograph, in presence of the pure standard.

The experiment used for alkaloids identification is called Direct infusion Mass Spectometry (DIMS). This method also if is not a pure chromatography, and it is extensively used in metabolomics (Kirwan, J., Weber, R., Broadhurst, D. et al. Direct infusion mass spectrometry metabolomics dataset: a benchmark for data processing and quality control. Sci Data 1, 140012 (2014). https://doi.org/10.1038/sdata.2014.12; J. Proteome Res. 2017, 16, 4, 1646–1658 Publication Date:February 28, 2017 https://doi.org/10.1021/acs.jproteome.6b01003).Moreover, DIMS is the gold standard to create data set and has the advantage of improving instrument sensitivity.(hristinaLooße, Sara Galozzi, Linde Debor, Mattijs K. Julsing, Bruno Bühler, Andreas Schmid, KatalinBarkovits, Thorsten Müller, KatrinMarcus,Direct infusion-SIM as fast and robust method for absolute protein quantification in complex samples, EuPA Open Proteomics,Volume 7,2015,Pages 20-26, ISSN 2212-9685,https://doi.org/10.1016/j.euprot.2015.03.001).

References were added to the text.

Minor:

  1. This error was well corrected in the manuscript.
  2. This error was well corrected in the manuscript.

Round 3

Reviewer 3 Report

Analytical part still needs explanation:

Authors: „This method also if is not a pure chromatography, and it is extensively used in metabolomics…”

Response:  Dear Authors, I do not question the method and I am aware that such a method is often applied (so the adding general references for the methodology is unnecessary). But the presented results raise doubts. Usually the obtained results (mass spectra) should be compared with literature or appropriate databases.

Masses of alkaloids given in Table 1 do not match to literature data, e.g. for Ergovaline parent [M-H]+ ion is 534.2716 (see e.g.  doi.org/10.1080/00498254.2018.1542187; A. Sobhani Najafabadi et al. / RPS 2010; 5(2): 136-143) and in Table 1 is 543.300;  [M-H]+  for Trisphaeridine in Table is 223.000 but mass of [M-H]+  ion for this compound is 224.0706… Check carefully whole Table. I think that the addition of literature (as a last column in Table 1) where  mass data for the particular  alkaloids were described (some examples are given above) or just full mass spectra for standards included as Supplementary material could clarify any doubts.

Authors: „in presence of the pure standard”

Response: No standards are mention in Materials and Methods section

Minor errors are still in the text:

Line 158: Unnecessary dot

Lines 6, 228: lack of space ambrosioides(L.), 500mg/kg

Line 16: “–The”

Author Response

Dear Reviewer 3,

We would like to thank Reviewer 3 for his/her thoughtful comments and the efforts made toward improving our manuscript. We have taken all the comments on board to improve and clarify the manuscript.

  • Table 1. Was revised and the selected mass-to-charge ratio results are as in the obtained mass spectra.
  • We have added to the supplementary file, the full mass spectra of standards as suggested (see Figure S3), to clear any doubts.

We hope this revision will satisfy reviewers’ queries, and that our work will be considered for publication in Toxins.

With kind regards

Round 4

Reviewer 3 Report

Dear Authors,

Still it is unknown on what basis the identification of components has been carried out. SIM (selected ion monitoring) mode is used to quantitative analysis when the composition of the sample is known. If literature data describing the chemical composition of the studied extract is available, such procedure is correct. However, in Table 1 no references are included. To identify unknown components of the sample, scan mode is necessary. Moreover, selected ions  (Table 1) do not correspond with (M-H+) ions. On what basis was appropriate ions selected to monitor target compound? If You cannot to provide mass spectrum, add references to justify the choice of ions monitored. 

The data shown in Figure S3 are not mass spectra. These are just extracted ions with specific mass. Moreover, no peak is visible on Fig. S3 A. It means that standard has no signal at chosen m/z.

"Pure standards used (...)" - Are they commercially available? No manufacturer of standards is given.

Biological experiments are interesting; so, if You cannot improve chemical investigation I suggest to remove this part from the manuscript.

Author Response

Dear Reviewer,

We highly thank you for your comments on our manuscript, we believe that your comments have helped us improve the quality of the analytical part. We have provided enough time to answer carefully your questions. You will find below the answer to each question.

Question 1 - On what basis was appropriate ions selected to monitor the target compound? If You cannot to provide mass spectrum, add references to justify the choice of ions monitored. 

Response 1 - The identification was firstly made in “Scan” mode, followed by the optimization of a SIM acquisition on the [M-H]+ peak found in the literature for each alkaloid and of all the peaks with the same behavior when the standards are injected at different concentrations. For example, when pure Demethylmaritidine was injected at the concentration of 25 ng/ml, the AUC of 274.33 was 15381 while that of 273.00 was 9874. When the standard was injected at 50 ng/ml AUC of 274.33 was 30754 while that of 273.00 was 19764. Such proportion was maintained for all the concentrations injected, by all the selected m/z compared to their [M-H]+ of reference.  The selected ions were chosen also to avoid interference of other components of the extract. The chosen peaks have the advantage of giving a better signal when the whole extract is injected. To demonstrate the quality of the investigation, the following table compares the literature [M-H]+ spectra and our chosen peaks, for pure standards (see the new supplementary file, Table S1).

Question 2 - The data shown in Figure S3 are not mass spectra. These are just extracted ions with specific mass. Moreover, no peak is visible on Fig. S3 A. It means that standard has no signal at chosen m/z.

Response 2 – Thank you for your observation, we have deleted Figure S3 and it was replaced with Table S1 (See the new supplementary file).

Question 3 - "Pure standards used (...)" - Are they commercially available? No manufacturer of standards is given.

Response 3 - Ergovaline, Galanthamie, Norgalanthamine, Anhydrolycorine, Trisphaeridine, Crinine, Demethylmaritidine, N-formylgalanthamine, and Peramine were purchased from Sigma Aldrich. We have added these informations to Materials and Methods section. Thank you.